# Boosting Robustness Certification of Neural Networks

**Gagandeep Singh, Timon Gehr, Markus Püschel, Martin Vechev**
Department of Computer Science
ETH Zurich, Switzerland
{gsingh,timon.gehr,pueschel,martin.vechev}@inf.ethz.ch

## ABSTRACT

We present a novel approach for the certification of neural networks against adversarial perturbations which combines scalable overapproximation methods with precise (mixed integer) linear programming. This results in significantly better precision than state-of-the-art verifiers on challenging feedforward and convolutional neural networks with piecewise linear activation functions.

## 1 INTRODUCTION

Neural networks are increasingly applied in critical domains such as autonomous driving (Bojarski et al., 2016), medical diagnosis (Amato et al., 2013), and speech recognition (Hinton et al., 2012). However, it has been shown by Goodfellow et al. (2014) that neural networks can be vulnerable against *adversarial* attacks, i.e., imperceptible input perturbations cause neural networks to misclassify. To address this challenge and prove that a network is free of adversarial examples (usually, in a region around a given input), recent work has started investigating the use of certification techniques. Current verifiers can be broadly classified as either complete or incomplete.

Complete verifiers are exact, i.e., if the verifier fails to certify a network then the network is non-robust (and vice-versa). Existing complete verifiers are based on Mixed Integer Linear Programming (MILP) (Lomuscio & Maganti, 2017; Fischetti & Jo, 2018; Dutta et al., 2018; Cheng et al., 2017) or SMT solvers (Katz et al., 2017; Ehlers, 2017). Although precise, these can only handle networks with a small number of layers and neurons. To scale, incomplete verifiers usually employ overapproximation methods and hence they are sound but may fail to prove robustness even if it holds. Incomplete verifiers use methods such as duality (Dvijotham et al., 2018), abstract interpretation (Gehr et al., 2018; Singh et al., 2018; 2019), linear approximations (Weng et al., 2018; Wong & Kolter, 2018; Zhang et al., 2018), semidefinite relaxations (Raghunathan et al., 2018), combination of linear and non-linear approximation (Xiang et al., 2017), or search space discretization (Huang et al., 2017). Incomplete verifiers are more scalable than complete ones, but can suffer from precision loss for deeper networks. In principle, incomplete verifiers can be made asymptotically complete by iteratively refining the input space (Wang et al., 2018a) or the neurons (Wang et al., 2018b); however, in the worst case, this may eliminate any scalability gains and thus defeat the purpose of using overapproximation in the first place.

**This work: boosting complete and incomplete verifiers.** A key challenge then is to design a verifier which improves the precision of incomplete methods and the scalability of complete ones. In this work, we make a step towards addressing this challenge based on two key ideas: (i) a combination of state-of-the-art overapproximation techniques used by incomplete methods, including LP relaxations, together with MILP solvers, often employed in complete verifiers; (ii) a novel heuristic, which points to neurons whose approximated bounds should be refined. We implemented these ideas in a system called *RefineZono*, and showed that is is faster than state-of-the-art complete verifiers on small networks while improving precision of existing incomplete verifiers on larger networks.

The recent works of (Wang et al., 2018b) and Tjeng et al. (2019) have also explored the combination of linear programming with overapproximation. However, both use simpler and coarser overapproximations than ours. Our evaluation shows that *RefineZono* is faster than both for complete verification. For example, *RefineZono* is faster than the work of Tjeng et al. (2019) for the complete

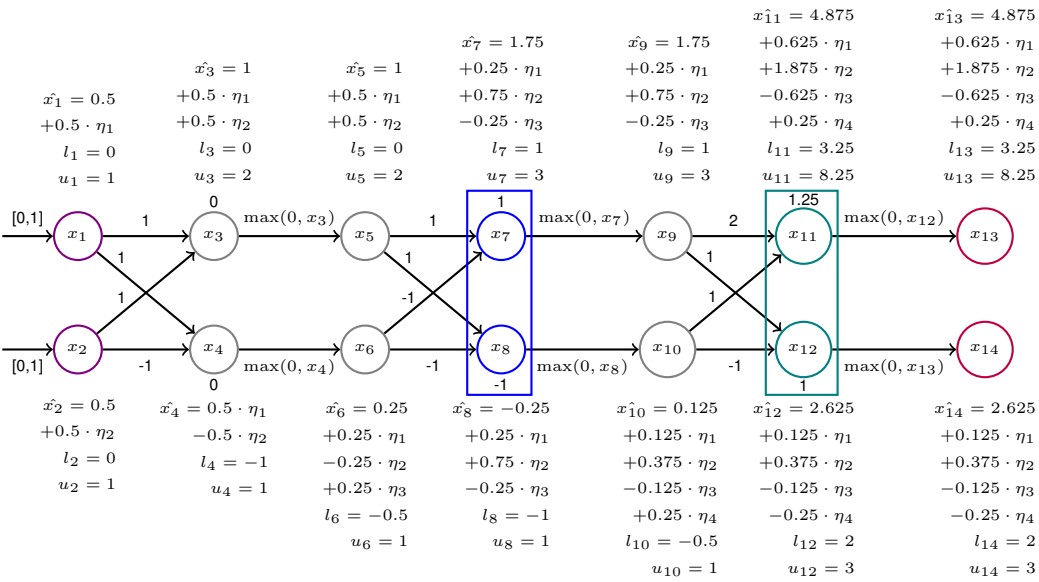

Figure 1: Robustness analysis of a toy example neural network using our method. Here, approximation results computed with *DeepZ* (blue box) are refined using MILP whereas those in green are refined using LP.

verification of a $3 \times 50$ network, while for the larger $9 \times 200$ network their method does not finish within multiple days on images which *RefineZono* verifies in $\approx 14$ minutes.

**Main contributions.** Our main contributions are:

- A refinement-based approach for certifying neural network robustness that combines the strengths of fast overapproximation methods with MILP solvers and LP relaxations.

- A novel heuristic for selecting neurons whose bounds should be further refined.

- A complete end-to-end implementation of our approach in a system called *RefineZono*, publicly available at https://github.com/eth-sri/eran.

- An evaluation, showing that *RefineZono* is more precise than existing state-of-the-art incomplete verifiers on larger networks and faster (while being complete) than complete verifiers on smaller networks.

## 2 OVERVIEW

We now demonstrate how our method improves the precision of a state-of-the-art incomplete verifier. The main objective here is to provide an intuitive understanding of our approach; full formal details are provided in the next section.

Consider the simple fully connected feedforward neural network with ReLU activations shown in Fig. 1. There are two inputs to the network, both in the range $[0, 1]$. The network consists of an input layer, two hidden layers, and one output layer. Each layer consist of two neurons each. For our explanation, we separate each neuron into two parts: one represents the output of the affine transformation while the other captures the output of the ReLU activation. The weights for the affine transformation are represented by weights on the edges. The bias for each node is shown above or below it. Our goal is to verify that for any input in $[0, 1] \times [0, 1]$, the output at neuron $x_{13}$ is greater than the output at $x_{14}$.

We now demonstrate how our verifier operates on this network. We assume that the analysis results after the second affine transformation are refined using a MILP formulation of the network whereas the results after the third affine transformation are refined by an LP formulation of the network. In

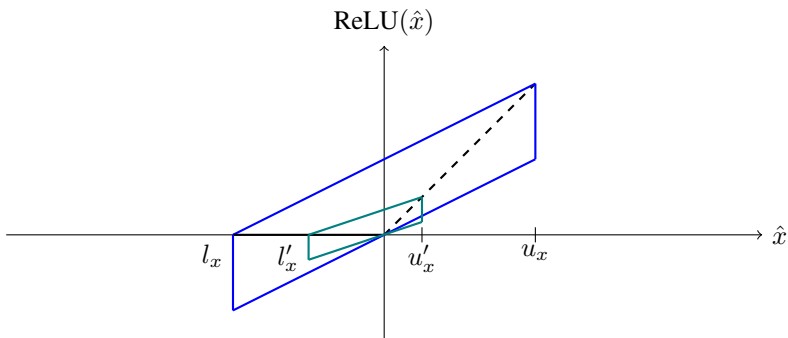

Figure 2: ReLU transformers, computing an affine form. Here, $l_x, u_x$ are the original bounds, whereas $l'_x, u'_x$ are the refined bounds. The slope of the two non-vertical parallel blue lines is $\lambda = u_x/(u_x - l_x)$ and the slope of the two non-vertical parallel green lines is $\lambda' = u'_x/(u'_x - l'_x)$. The blue parallelogram is used to compute an affine form in *DeepZ*, whereas the green parallelogram is used to compute the output of the refined ReLU transformer considered in this work.

the next section, we will explain our heuristic for selecting MILP or LP formulations of different neurons in the network. Our analysis leverages the Zonotope domain (Ghorbal et al., 2009) together with the abstract Zonotope transformers specialized to neural network activations as used in *DeepZ* (Singh et al., 2018), a state of the art verifier for neural network robustness. The Zonotope domain associates an affine form $\hat{x}$ with each neuron $x$ in the network:

$$\hat{x} := c_0 + \sum_{i=1}^{p} c_i \cdot \eta_i$$

Here, $c_0, c_i \in \mathbb{R}$ are real coefficients and $\eta_i \in [s_i, t_i] \subseteq [-1, 1]$ are the noise symbols, which are shared between the affine forms for different neurons. This sharing makes the domain relational and thus more precise than non-relational domains such as Interval (Box). An abstract element in our analysis is an intersection between a Zonotope (given as a list of affine forms) and a bounding box. Thus, for each neuron $x$, we keep the affine form $\hat{x}$ and an interval $[l_x, u_x]$.

**First layer.** Our analysis starts by setting

$$\hat{x}_1 = 0.5 + 0.5 \cdot \eta_1, l_1 = 0, u_1 = 1$$

and

$$\hat{x}_2 = 0.5 + 0.5 \cdot \eta_2, l_2 = 0, u_2 = 1,$$

representing the input $[0, 1]$ at $x_1$ and $[0, 1]$ at $x_2$ in our domain, respectively. Next, an affine transformation is applied on the inputs resulting in the output

$$\hat{x}_3 = \hat{x}_1 + \hat{x}_2 = 1 + 0.5 \cdot \eta_1 + 0.5 \cdot \eta_2, l_3 = 0, u_3 = 2$$

and

$$\hat{x}_4 = \hat{x}_1 - \hat{x}_2 = 0.5 \cdot \eta_1 - 0.5 \cdot \eta_2, l_4 = -1, u_4 = 1.$$

Note that the Zonotope affine transformer is *exact* for this transformation. Next, the Zonotope ReLU transformer is applied. We note that as $l_3 \geq 0$, the neuron $x_3$ provably takes only non-negative values. Thus, the ReLU Zonotope transformer outputs $\hat{x}_5 = \hat{x}_3$ and we set $l_5 = l_3, u_5 = l_3$ which is the exact result. For $x_4$, $l_4 < 0$ and $u_4 > 0$ and thus neuron $x_4$ can take both positive and negative values. The corresponding output does not have a closed affine form and hence the approximation in blue shown in Fig. 2 is used to compute the result. This approximation minimizes the area of the result in the input-output plane and introduces a new noise symbol $\eta_3 \in [-1, 1]$. The result is

$$\hat{x}_6 = 0.25 + 0.25 \cdot \eta_1 - 0.25 \cdot \eta_2 + 0.25 \cdot \eta_3, l_6 = -0.5, u_6 = 1.$$

Note that the Zonotope approximation for $x_6$ from Fig. 2 permits negative values whereas $x_6$ can only take non-negative values in the concrete. This overapproximation typically accumulates as the analysis progresses deeper into the network, resulting in overall imprecision and failure to prove properties that actually hold.

**MILP-based refinement at second layer.** Next, the analysis handles the second affine transformation and computes

$$\hat{x}_7 = \hat{x}_5 - \hat{x}_6 + 1 = 1.75 + 0.25 \cdot \eta_1 + 0.75 \cdot \eta_2 - 0.25 \cdot \eta_3, l_7 = 0.5, u_7 = 3$$

and

$$\hat{x}_8 = \hat{x}_5 - \hat{x}_6 - 1 = -0.25 + 0.25 \cdot \eta_1 + 0.75 \cdot \eta_2 - 0.25 \cdot \eta_3, l_8 = -1.5, u_8 = 1.$$

Here, $x_7$ is provably positive, whereas $x_8$ can take both positive and negative values. Due to the approximation for $x_6$, the bounds for $x_7$ and $x_8$ are imprecise. Note that the *DeepZ* ReLU transformer for $x_8$ applied next will introduce more imprecision and although the ReLU transformer for provably positive inputs such as $x_7$ does not lose precision with respect to the input, it still propagates the imprecision in the computation of the abstract values for $x_7$.

Thus, to reduce precision loss, in our method we refine the bounds for both $x_7$ and $x_8$ by formulating the network up to (and including) the second affine transformation as a MILP instance based on a formulation from Tjeng et al. (2019) and compute bounds for $x_7$ and $x_8$, respectively. The MILP solver improves the lower bounds for $x_7$ and $x_8$ to 1 and $-1$, respectively, which then updates the corresponding lower bounds in our abstraction, i.e., $l_7 = 1$ and $l_8 = -1$.

Next, the ReLU transformer is applied. Since $x_7$ is provably positive, we get $\hat{x}_9 = \hat{x}_7, l_9 = l_7$, and $u_9 = u_7$. We note that $x_8$ can take both positive and negative values and is therefore approximated. However, the ReLU transformer now uses the refined bounds instead of the original bounds and thus the approximation shown in green from Fig. 2 is used. This approximation has smaller area in the input-output plane compared to the blue one and thus reduces the approximation error. The result is

$$\hat{x}_{10} = 0.125 + 0.125 \cdot \eta_1 + 0.375 \cdot \eta_2 - 0.125 \cdot \eta_3 + 0.25 \cdot \eta_4, l_8 = -0.5, u_8 = 1.$$

**LP-based refinement at final layer.** Continuing with the analysis, we now process the final affine transformation, which yields

$$\hat{x}_{11} = 4.875 + 0.625 \cdot \eta_1 + 1.875 \cdot \eta_2 - 0.625 \cdot \eta_3 + 0.25 \cdot \eta_4, l_{11} = 1.75, u_{11} = 8.25$$

and

$$\hat{x}_{12} = 2.625 + 0.125 \cdot \eta_1 + 0.375 \cdot \eta_2 - 0.125 \cdot \eta_3 - 0.25 \cdot \eta_4, l_{12} = 1.75, u_{12} = 3.5.$$

Due to the approximations from previous layers, the computed values can be imprecise. We note that, as the analysis proceeds deeper into the network, refining bounds with MILP becomes expensive. Thus, we refine the bounds by encoding the network up to (and including) the third affine transformation using the faster LP relaxation of the network based on Ehlers (2017) and compute the bounds for $x_{11}$ and $x_{12}$, respectively. This leads to better results for $l_{11} = 3.25$, $l_{12} = 2$, and $u_{12} = 3$. As both $x_{11}$ and $x_{12}$ are provably positive, the subsequent ReLU transformations set $\hat{x}_{13} = \hat{x}_{11}, l_{13} = l_{11}, u_{13} = u_{11}$ and $\hat{x}_{14} = \hat{x}_{12}, l_{14} = l_{12}, u_{14} = u_{12}$.

**Proving robustness.** Since the lower bound $l_{13}$ for $x_{13}$ is greater than the upper bound $u_{14}$ for $x_{14}$, our analysis can prove that the given neural network provides the same label for all inputs in $[0, 1] \times [0, 1]$ and is thus robust. In contrast, *DeepZ* without our refinement would compute

$$\hat{x}_{13} = 4.95 + 0.6 \cdot \eta_1 + 1.8 \cdot \eta_2 - 0.6 \cdot \eta_3 + 0.3 \cdot \eta_4, l_{13} = 1.65, u_{13} = 8.25$$

and

$$\hat{x}_{14} = 2.55 + 0.15 \cdot \eta_1 + 0.45 \cdot \eta_2 - 0.15 \cdot \eta_3 - 0.3 \cdot \eta_4, l_{14} = 1.5, u_{14} = 3.6.$$

As a result, *DeepZ* fails to prove that $x_{13}$ is greater than $x_{14}$, and thus fails to prove robustness.

**Generalization to other abstractions.** We note that our refinement-based approach is not restricted to the Zonotope domain. It can be extended for refining the results computed by other abstractions such as Polyhedra (Singh et al., 2017) or the abstraction used in *DeepPoly* (Singh et al., 2019). For example, the ReLU transformer in Singh et al. (2019) also depends on the bounds of input neurons and thus it will benefit from the precise bounds computed using our refinement. Since *DeepPoly* often produces more precise results than *DeepZ*, we believe a combination of this work with *DeepPoly* will further improve verification results.

## 3 OUR APPROACH

We now describe our approach in more formal terms. As in the previous section, we will consider affine transformations and ReLU activations as separate layers. As illustrated earlier, the key idea will be to combine abstract interpretation (Cousot & Cousot, 1977) with exact and inexact MILP formulations of the network, which are then solved, in order to compute more precise results for neuron bounds. We begin by describing the core ingredients of abstract interpretation.

Our approach requires an abstract domain $\mathbb{A}_n$ over $n$ variables (i.e., some set whose elements can be encoded symbolically) such as Interval, Zonotope, the abstraction in *DeepPoly*, or Polyhedra. An abstract domain has a bottom element $\perp \in \mathbb{A}_n$ as well as the following components:

- A (potentially non-computable) concretization function $\gamma_n \colon \mathbb{A}_n \to \mathcal{P}(\mathbb{R}^n)$ that associates with each abstract element $a \in \mathbb{A}_n$ the set of concrete points from $\mathbb{R}^n$ that it abstracts. We have $\gamma_n(\perp) = \emptyset$.

- An abstraction function $\alpha_n \colon \mathbb{B}_n \to \mathbb{A}_n$, where $\mathbb{X} \subseteq \gamma_n(\alpha_n(\mathbb{X}))$ for all $\mathbb{X} \in \mathbb{B}_n$. We assume that $\alpha_n(\prod_i [l_i, u_i])$ is a computable function of $\boldsymbol{l}, \boldsymbol{u} \in \mathbb{R}^n$. Here, $\mathbb{B}_n = \bigcup_{\boldsymbol{l}, \boldsymbol{u} \in \mathbb{R}^n} \prod_i [l_i, u_i]$ and $\prod_i [l_i, u_i] = \{\boldsymbol{x} \in \mathbb{R}^n \mid l_i \leq x_i \leq u_i\}$. (For many abstract domains, $\alpha_n$ can be defined on a larger domain $\mathbb{B}_n$, but in this work, we only consider Interval input regions.)

- A bounding box function $\iota_n \colon \mathbb{A}_n \to \mathbb{R}^n \times \mathbb{R}^n$, where $\gamma_n(a) \subseteq \prod_i [l_i, u_i]$ for $(\boldsymbol{l}, \boldsymbol{u}) = \iota_n(a)$ for all $a \in \mathbb{A}_n$.

- A meet operation $a \sqcap L$ for each $a \in \mathbb{A}_n$ and linear constraints $L$ over $n$ real variables, where $\{x \in \gamma_n(a) \mid L(x)\} \subseteq \gamma_n(a \sqcap L)$.

- An affine abstract transformer $T^{\#}_{\boldsymbol{x} \mapsto \boldsymbol{A}\boldsymbol{x} + \boldsymbol{b}} \colon \mathbb{A}_m \to \mathbb{A}_n$ for each transformation of the form $(\boldsymbol{x} \mapsto \boldsymbol{A}\boldsymbol{x} + \boldsymbol{b}) \colon \mathbb{R}^m \to \mathbb{R}^n$, where
$$\{\boldsymbol{A}\boldsymbol{x} + \boldsymbol{b} \mid \boldsymbol{x} \in \gamma_n(a)\} \subseteq \gamma_n(T^{\#}_{\boldsymbol{x} \mapsto \boldsymbol{A}\boldsymbol{x} + \boldsymbol{b}}(a))$$
for all $a \in \mathbb{A}_m$.

- A ReLU abstract transformer $T^{\#}_{\mathrm{ReLU}|_{\prod_i [l_i, u_i]}} \colon \mathbb{A}_n \to \mathbb{A}_n$, where
$$\{\mathrm{ReLU}(\boldsymbol{x}) \mid \boldsymbol{x} \in \gamma_n(a) \cap \prod_i [l_i, u_i]\} \subseteq T^{\#}_{\mathrm{ReLU}|_{\prod_i [l_i, u_i]}}(a)$$
for all abstract elements $a \in \mathbb{A}_n$ and for all lower and upper bounds $\boldsymbol{l}, \boldsymbol{u} \in \mathbb{R}^n$ on input activations of the ReLU operation.

**Verification via Abstract interpretation.** As first shown by Gehr et al. (2018), any such abstract domain induces a method for robustness certification of neural networks with ReLU activations.

For example, assume that we want to certify that a given neural network $f \colon \mathbb{R}^m \to \mathbb{R}^n$ considers class $i$ more likely than class $j$ for all inputs $\bar{\boldsymbol{x}}$ with $||\bar{\boldsymbol{x}} - \boldsymbol{x}||_\infty \leq \epsilon$ for a given $\boldsymbol{x}$ and $\epsilon$. We can first use the abstraction function $\alpha_m$ to compute a symbolic overapproximation of the set of possible inputs $\bar{\boldsymbol{x}}$, namely
$$a_{\mathrm{in}} = \alpha_m(\{\bar{\boldsymbol{x}} \in \mathbb{R}^m \mid ||\bar{\boldsymbol{x}} - \boldsymbol{x}||_\infty \leq \epsilon\}).$$

Given that the neural network can be written as a composition of affine functions and ReLU layers, we can then propagate the abstract element $a_{\mathrm{in}}$ through the corresponding abstract transformers to obtain a symbolic overapproximation $a_{\mathrm{out}}$ of the concrete outputs of the neural network.

For example, if the neural network $f(\boldsymbol{x}) = \boldsymbol{A}' \cdot \mathrm{ReLU}(\boldsymbol{A}\boldsymbol{x} + \boldsymbol{b}) + \boldsymbol{b}'$ has a single hidden layer with $h$ hidden neurons, we first compute $a' = T^{\#}_{\boldsymbol{x} \mapsto \boldsymbol{A}\boldsymbol{x} + \boldsymbol{b}}(a_{\mathrm{in}})$, which is a symbolic overapproximation of the input to the ReLU activation function. We then compute $(\boldsymbol{l}, \boldsymbol{u}) = \iota_h(a')$ to obtain opposite corners of a bounding box of all possible ReLU input activations, such that we can apply the ReLU abstract transformer:
$$a'' = T^{\#}_{\mathrm{ReLU}|_{\prod_i [l_i, u_i]}}(a').$$

Finally, we apply the affine abstract transformer again to obtain $a_{\mathrm{out}} = T^{\#}_{\boldsymbol{x} \mapsto \boldsymbol{A}'\boldsymbol{x} + \boldsymbol{b}'}(a'')$. Using our assumptions, we can conclude that the set $\gamma_n(a_{\mathrm{out}})$ contains all output activations that $f$ can possibly produce when given any of the inputs $\bar{\boldsymbol{x}}$. Therefore, if $a_{\mathrm{out}} \sqcap (x_i \leq x_j) = \perp$, we have proved the property: for all $\bar{\boldsymbol{x}}$, the neural network considers class $i$ more likely than class $j$.

**Incompleteness.** While this approach is sound (i.e., whenever we prove the property, it actually holds), it is incomplete (i.e., we might not prove the property, even if it holds), because the abstract transformers produce a superset of the set of concrete outputs that the corresponding concrete executions produce. This can be quite imprecise for deep neural networks, because the overapproximations introduced in each layer accumulate.

**Refining the bounds.** To combat spurious overapproximation, we use mixed integer linear programming (MILP) to compute refined lower and upper bounds $l', u'$ after applying each affine abstract transformer (except for the first layer). We then refine the abstract element using the meet operator of the underlying abstract domain and the linear constraints $l'_i \leq x_i \leq u'_i$ for all input activations $i$, i.e., we replace the current abstract element $a$ by $a' = a \sqcap (\bigwedge_i l'_i \leq x_i \leq u'_i)$, and continue analysis with the refined abstract element.

Importantly, we obtain a more refined abstract transformer for ReLU than the one used in *DeepZ* by leveraging the new lower and upper bounds. That is, using the tighter bounds $l'_x, u'_x$ for $x$, we define the ReLU transformer for $y := \max(0, x)$ as follows:

$$\hat{y} = \begin{cases} \hat{x}, & \text{if } l'_x > 0, \\ 0, & \text{if } u'_x \leq 0, \\ \lambda \cdot \hat{x} + \mu + \mu \cdot \epsilon_{\text{new}}, & \text{otherwise.} \end{cases}$$

Here $\lambda = \frac{u'_x}{u'_x - l'_x}$, $\mu = -\frac{u'_x \cdot l'_x}{2 \cdot (u'_x - l'_x)}$, and $\epsilon_{\text{new}} \in [-1, 1]$ is a new noise symbol.

The refined ReLU transformer benefits from the improved bounds. For example, when $l_x < 0$ and $u_x > 0$ holds for the original bounds then after refinement:

- If $l'_x > 0$, then the output is the same as the input and no overapproximation is added.
- Else if $u'_x \leq 0$, then the output is exact.
- Otherwise, as shown in Fig. 2, the approximation with the tighter $l'_x$ and $u'_x$ has smaller area in the input-output plane than the original transformer that uses the imprecise $l_x$ and $u_x$.

**Obtaining constraints for refinement.** To enable refinement with MILP, we need to obtain constraints which fully capture the behavior of the neural network up to the last layer whose abstract transformer has been executed. In our encoding, we have one variable for each neuron and we write $x_i^{(k)}$ to denote the variable corresponding to the activation of the $i$-th neuron in the $k$-th layer, where the input layer has $k = 0$. Similarly, we write $l_i^{(k)}$ and $u_i^{(k)}$ to denote the best derived lower and upper bounds for this neuron.

From the input layer, we obtain constraints of the form $l_i^0 \leq x_i^{(0)} \leq u_i^0$, from affine layers, we obtain constraints of the form $x_i^{(k)} = \sum_j a_{ij}^{(k-1)} x_j^{(k-1)} + b_i^{(k-1)}$ and from ReLU layers we obtain constraints of the form $x_i^{(k)} = \max(0, x_i^{(k-1)})$.

**MILP.** Let $\varphi^{(k)}$ denote the conjunction of all constraints up to and including those from layer $k$. To obtain the best possible lower and upper bounds for layer $k$ with $p$ neurons, we need to solve the following $2 \cdot p$ optimization problems:

$$l_i'^{(k)} = \min_{\substack{x_1^{(0)}, \dots, x_p^{(k)} \\ \text{s.t. } \varphi^{(k)}(x_1^{(0)}, \dots, x_p^{(k)})}} x_i^{(k)}, \text{ for } i = 1, \dots, p,$$

$$u_i'^{(k)} = \max_{\substack{x_1^{(0)}, \dots, x_p^{(k)} \\ \text{s.t. } \varphi^{(k)}(x_1^{(0)}, \dots, x_p^{(k)})}} x_i^{(k)}, \text{ for } i = 1, \dots, p.$$

As was shown by Tjeng et al. (2019), such optimization problems can be encoded exactly as MILP instances using the bounds computed by abstract interpretation and the instances can then be solved using off-the-shelf MILP solvers to compute $l_i'^{(k)}$ and $u_i'^{(k)}$.

**LP relaxation.** While not introducing any approximation, unfortunately, current MILP solvers do not scale to larger neural networks. It becomes increasingly more expensive to refine bounds with the MILP-based formulation as the analysis proceeds deeper into the network. However, for soundness it is not crucial that the produced bounds are the best possible: for example, plain abstract interpretation uses sound bounds produced by the bounding box function $\iota$ instead. Therefore, for deeper layers in the network, we explore the trade-off between precision and scalability by also considering an intermediate method, which is faster than exact MILP, but also more precise than abstract interpretation. We relax the constraints in $\varphi^{(k)}$ using the bounds computed by abstract interpretation in the same way as Ehlers (2017) to obtain a set of weaker linear constraints $\varphi_{\mathrm{LP}}^{(k)}$. We then use the solver to solve the relaxed optimization problems that are constrained by $\varphi_{\mathrm{LP}}^{(k)}$ instead of $\varphi^{(k)}$, producing possibly looser bounds $\boldsymbol{l}'^{(k)}$ and $\boldsymbol{u}'^{(k)}$. Note that the encoding of subsequent layers depends on the bounds computed in previous layers, where tighter bounds reduce the amount of newly introduced approximation.

**Anytime MILP relaxation.** MILP solvers usually provide the option to provide an explicit time-out after which the solver must terminate. In return, the solver may not be able to solve the instance exactly, but it will instead provide lower and upper bounds on the objective function in a best-effort fashion. This provides another way to compute sound but inexact bounds $\boldsymbol{l}'^{(k)}$ and $\boldsymbol{u}'^{(k)}$.

In practice, we choose a fraction $\theta \in (0,1]$ of neurons in a given layer $k$ and compute bounds for them using MILP with a timeout $T$ in a first step. In the second step, for a fraction $\delta \in [0, 1-\theta]$ of neurons in the layer, we set the timeout to $\beta \cdot \overline{T}$, where $\overline{T}$ is the average time taken by the MILP solver to solve one of the instances from the first step and $\beta \in [0,1]$ is a parameter.

**Neuron selection heuristic.** To select the $\theta$-fraction of neurons for the first step of the anytime MILP relaxation for the $k$-th layer, we rank the neurons. If the next layer is a ReLU layer, we first ignore all neurons whose activations can be proven to be non-positive using abstract interpretation (i.e., using the bounds produced by $\iota$), because in this case it is already known that ReLU will map the activation to $0$. The remaining neurons are ordered in up to two different ways, once by width (i.e. neuron $i$ has key $u_i^{(k)} - l_i^{(k)}$), and possibly once by the sum of absolute output weights. i.e., if the next layer is a fully connected layer $\boldsymbol{x} \mapsto \boldsymbol{Ax} + \boldsymbol{b}$, the key of neuron $i$ is $\sum_j |A_{i,j}|$. If the next layer is a ReLU layer, we skip the ReLU layer and use the weights from the fully connected layer that follows it (if any). The two ranks of a neuron in both orders are added, and the $\theta$-fraction with smallest rank sum is selected and their bounds are refined with a timeout of $T$ whereas the next $\delta$-fraction of neurons are refined with a timeout of $\beta \cdot \overline{T}$.

***RefineZono*: end-to-end approach.** To certify robustness of deep neural networks, we combine MILP, LP relaxation, and abstract interpretation. We first pick numbers of layers $k_{\mathrm{MILP}}, k_{\mathrm{LP}}, k_{\mathrm{AI}}$ that sum to the total number of layers of the neural network. For the analysis of the first $k_{\mathrm{MILP}}$ layers, we refine bounds using anytime MILP relaxation with the neuron selection heuristic. As an optimization, we do not perform refinement after the abstract transformer for the first layer in case it is an affine transformation, as the abstract domain computes the tightest possible bounding box for an affine transformation of a box (this is always the case in our experiments). For the next $k_{\mathrm{LP}}$ layers, we refine bounds using LP relaxation (i.e., the network up to the layer to be refined is encoded using linear constraints) combined with the neuron selection heuristic. For the remaining $k_{\mathrm{AI}}$ layers, we use abstract interpretation without additional refinement (however, this also benefits from refinement that was performed in previous layers), and compute the bounds using $\iota$.

**Final property certification.** Let $k$ be the index of the last layer and $p$ be the number of output classes. We can encode the final certification problem using the output abstract element $a_{\mathrm{out}}$ obtained after applying the abstract transformer for the last layer in the network. If we want to prove that class $i$ is assigned a higher probability than class $j$, it suffices to show that $a_{\mathrm{out}} \sqcap (x_i^{(k)} \leq x_j^{(k)}) = \bot$. If this fails, one can resort to complete verification using MILP: the property is satisfied if and only if the set of constraints $\varphi^{(k)}(x_1^{(0)}, \dots, x_p^{(k)}) \wedge (x_i^{(k)} \leq x_j^{(k)})$ is unsatisfiable.

Table 1: Neural network architectures used in our experiments.

| Dataset | Model | Type | #Neurons | #layers | Defense |
|---------|-------|------|----------|---------|---------|
| MNIST | $3 \times 50$ | fully connected | 160 | 3 | None |
| | $5 \times 100$ | fully connected | 510 | 5 | DiffAI |
| | $6 \times 100$ | fully connected | 610 | 6 | None |
| | $9 \times 100$ | fully connected | 910 | 9 | None |
| | $6 \times 200$ | fully connected | 1 210 | 6 | None |
| | $9 \times 200$ | fully connected | 1 810 | 9 | None |
| | ConvSmall | convolutional | 3 604 | 3 | None |
| | ConvBig | convolutional | 34 688 | 6 | DiffAI |
| | ConvSuper | convolutional | 88 500 | 6 | DiffAI |
| CIFAR10 | $6 \times 100$ | fully connected | 610 | 6 | None |
| | ConvSmall | convolutional | 4 852 | 3 | DiffAI |
| ACAS Xu | $6 \times 50$ | fully connected | 305 | 6 | None |

## 4 EVALUATION

We evaluate the effectiveness of our approach for the robustness verification of ReLU-based feedforward and convolutional neural networks. The results show that our approach enables faster complete verification than the state-of-the-art complete verifiers: Wang et al. (2018b) and Tjeng et al. (2019), and produces more precise results than state-of-the-art incomplete verifiers: *DeepZ* (Singh et al., 2018) and *DeepPoly* (Singh et al., 2019), when complete certification becomes infeasible.

We implemented our approach in a system called *RefineZono*. *RefineZono* uses Gurobi (Gurobi Optimization, LLC, 2018) for solving MILP and LP instances and is built on top of the ELINA library (eli, 2018; Singh et al., 2017) for numerical abstract domains. All of our code, neural networks, and images used in our experiments are publicly available at https://github.com/eth-sri/eran.

**Evaluation datasets.** We used the popular MNIST (Lecun et al., 1998), CIFAR10 (Krizhevsky, 2009), and ACAS Xu (Julian et al., 2018) datasets in our experiments. MNIST contains grayscale images of size $28 \times 28$ pixels whereas CIFAR10 contains RGB images of size $32 \times 32$. ACAS Xu contains 5 inputs representing aircraft sensor data.

**Neural networks.** Table 1 shows 12 different MNIST, CIFAR10, and ACAS Xu feedforward (FNNs) and convolutional networks (CNNs) with ReLU activations used in our experiments. Out of these 4 were trained to be robust against adversarial attacks using DiffAI (Mirman et al., 2018) whereas the remaining 8 had no adversarial training. The largest network in our experiments contains $> 88K$ neurons whereas the deepest network contains 9 layers.

**Robustness properties.** For MNIST and CIFAR10, we consider the $L_\infty$-norm (Carlini & Wagner, 2017) based adversarial region parameterized by $\epsilon \in \mathbb{R}$. Our goal here is to certify that the network produces the correct label on all points in the adversarial region. For ACAS Xu, our goal is to verify that the property $\phi_9$ (Katz et al., 2017) holds for the $6 \times 50$ network (known to be hard).

**Experimental setup.** All experiments for the $3 \times 50$ MNIST FNN and all CNNs were carried out on a 2.6 GHz 14 core Intel Xeon CPU E5-2690 with 512 GB of main memory; the remaining FNNs were evaluated on a 3.3 GHz 10 Core Intel i9-7900X Skylake CPU with a main memory of 64 GB.

**Benchmarks.** For each MNIST and CIFAR10 network, we selected the first 100 images from the respective test set and filtered out those images that were not classified correctly. We consider complete certification with *RefineZono* on the ACAS Xu network and the $3 \times 50$ MNIST network. For the $3 \times 50$ network, we choose an $\epsilon$ for which the incomplete verifier *DeepZ* certified $< 40\%$ of all candidate images. We consider incomplete certification for the remaining networks and choose an $\epsilon$ for which complete certification with *RefineZono* becomes infeasible.

Table 2: Precision and runtime of *RefineZono* vs. *DeepZ* and *DeepPoly*.

| Dataset | Model | $\epsilon$ | *DeepZ* | | *DeepPoly* | | *RefineZono* | |
|---|---|---|---|---|---|---|---|---|
| | | | precision(%) | time(s) | precision(%) | time(s) | precision(%) | time(s) |
| MNIST | $5 \times 100$ | 0.07 | 38 | 0.6 | 53 | 0.3 | 53 | 381 |
| | $6 \times 100$ | 0.02 | 31 | 0.6 | 47 | 0.2 | 67 | 194 |
| | $9 \times 100$ | 0.02 | 28 | 1.0 | 44 | 0.3 | 59 | 246 |
| | $6 \times 200$ | 0.015 | 13 | 1.8 | 32 | 0.5 | 39 | 567 |
| | $9 \times 200$ | 0.015 | 12 | 3.7 | 30 | 0.9 | 38 | 826 |
| | ConvSmall | 0.12 | 7 | 1.4 | 13 | 6.0 | 21 | 748 |
| | ConvBig | 0.2 | 79 | 7 | 78 | 61 | 80 | 193 |
| | ConvSuper | 0.1 | 97 | 133 | 97 | 400 | 97 | 665 |
| CIFAR10 | $6 \times 100$ | 0.0012 | 31 | 4.0 | 46 | 0.6 | 46 | 765 |
| | ConvSmall | 0.03 | 17 | 5.8 | 21 | 20 | 21 | 550 |

### 4.1 COMPLETE CERTIFICATION

*RefineZono* first runs *DeepZ* analysis on the whole network collecting the bounds for all neurons in the network. If *DeepZ* fails to certify the network, then the collected bounds are used to encode the robustness certification as a MILP instance (discussed in section 3).

**ACAS Xu** $6 \times 50$ **network.** As this network has only 5 inputs, we uniformly split the pre-condition defined by $\phi_9$ to produce $6\,300$ smaller input regions. We certify that the post-condition defined by $\phi_9$ holds for each region with *RefineZono*. *RefineZono* certifies that $\phi_9$ holds for the network in 227 seconds which is $> 4$x faster than the fastest verifier for ACAS Xu from Wang et al. (2018b).

**MNIST** $3 \times 50$ **network.** We use $\epsilon = 0.03$ for the $L_\infty$-norm attack. We compare *RefineZono* against the state-of-the-art complete verifier for MNIST from Tjeng et al. (2019). This approach is also MILP-based like ours, but it uses Interval analysis and LP to determine neuron bounds. We implemented the Interval analysis and LP-based analysis to determine the initial bounds. We call the MILP solver only if LP analysis (or Interval analysis) fails to certify. All complete verifiers certify the neural network to be robust against $L_\infty$-norm perturbations on $85\%$ of the images. The average runtime of *RefineZono*, MILP with bounds from the Interval analysis, and MILP with bounds from the LP analysis are 28, 123, and 35 seconds respectively. Based on our result, we believe that the Zonotope analysis offers a good middle ground between the speed of the Interval analysis and the precision of LP for bound computation, as it produces precise bounds faster than LP.

### 4.2 INCOMPLETE CERTIFICATION

We next compare *RefineZono* against *DeepZ* and *DeepPoly* for the incomplete robustness certification of the remaining networks. We note that *DeepZ* has the same precision as Fast-Lin (Weng et al., 2018) and *DeepPoly* has the same precision as CROWN (Zhang et al., 2018). The $\epsilon$ values used for the $L_\infty$-norm attack are shown in Table 2. The $\epsilon$ values for networks trained to be robust are larger than for networks that are not. For each verifier, we report the average runtime per image in seconds and the precision measured by the % of images for which the verifier certified the network to be robust. We note that running the Interval analysis to obtain initial bounds is too imprecise for these large networks with the $\epsilon$ values considered in our experiments. As a result, the approach from Tjeng et al. (2019) has to rely on applying LP per neuron to obtain precise bounds for the MILP solver which does not scale. For example, on the $9 \times 200$ network, determining bounds with LP already takes $> 20$ minutes (without calling the MILP solver which is more expensive than LP) whereas *RefineZono* has an average running time of $\approx 14$ minutes.

**Parameter values.** We experimented with different values of the analysis parameters $k_{\mathrm{MILP}}, k_{\mathrm{LP}}$, $k_{\mathrm{AI}}, \theta, \delta, \beta, T$ and chose values that offered the best tradeoff between performance and precision for the certification of each neural network. We refine the neuron bounds after all affine transformations

that are followed by a ReLU except the first one. In a given layer, we consider all neurons that can take positive values after the affine transformation as refinement candidates.

For the MNIST FNNs, we refine the bounds of the candidate neurons in layers 2-4 with MILP and those in the remaining layers using LP. For MILP based refinement, we use $\theta = \frac{\omega}{5^{k-2} \cdot p}$ where $\omega$ is the number of candidates and $p$ is the total number of neurons in layer $k$. For LP based refinement, we use $\theta = \frac{\omega}{2^{k-5} \cdot p}$. We use timeout $T = 1$ second, $\beta = 0.5$, and $\delta = \frac{\omega}{p} - \theta$ for both MILP and LP based refinements. For the CIFAR10 FNN, we use the same values except that we use $\theta = \frac{\omega}{2^{k-2} \cdot p}$ for MILP refinement and set $T = 6$ seconds for both MILP and LP based refinement as it is more expensive to refine neuron bounds in CIFAR10 networks due to these having more input neurons.

For the CNNs, the convolutional layers have large number of candidates so we do not refine these. Instead, we refine all candidates in the fully connected layers with a larger timeout so to compensate for the more difficult problem instances for the solver. For the MNIST ConvSmall, ConvBig and CIFAR10 ConvSmall networks, we refine all the candidate neurons using MILP with $T = 10$ seconds. For the MNIST ConvSuper network, we refine similarly but use LP with $T = 15$ seconds.

**Results for incomplete certification.**    Table 2 shows the precision and the average runtime of all three verifiers. *RefineZono* either improves or achieves the precision of the state-of-the-art verifiers on all neural networks. It certifies more images than *DeepZ* on all networks except the MNIST ConvSuper network. This is because *DeepZ* is already very precise for the $\epsilon$ considered. We could not try larger $\epsilon$ for this network, as the *DeepZ* analysis becomes too expensive. *RefineZono* certifies the network to be more robust on more images than *DeepPoly* on 6 out of 10 networks.

It can be seen that the number of neurons in the network is not the determining factor for the average runtime of *RefineZono*. We observe that *RefineZono* runs faster on the networks trained to be robust and the top three networks with the largest runtime for *RefineZono* are all networks not trained to be robust. This is because robust networks are relatively easier to certify and produce only a small number of candidate neurons for refinement, which are easier to refine by the solver. For example, even though the same parameter values are used for refining the results on the MNIST ConvSmall and ConvBig networks, the average runtime of *RefineZono* on the robustly trained ConvBig network with $\approx 35$K neurons, 6 layers and a perturbation region defined using $\epsilon = 0.2$ is almost 4 times less than on the non-robust ConvSmall network with only $3\,604$ neurons, 3 layers and a smaller $\epsilon = 0.12$.

### 4.3    EFFECT OF NEURON SELECTION HEURISTIC

We use the neuron selection heuristic from section 3 to determine neurons which need to be refined more than others for FNNs, as refining all neurons in a layer with MILP can significantly slow down the analysis. To check whether our heuristic can identify important neurons, we ran the analysis on the MNIST $9 \times 200$ FNN by keeping all analysis parameters the same, except instead of selecting the neurons with the smallest rank sum first we selected the neurons with the largest rank sum first (thus refining neurons more if our heuristic deems them unimportant). With this change, the average runtime does not change significantly. However, the modified analysis loses precision and fails to certify two images that the analysis refining with our neuron selection heuristic succeeds on.

## 5    CONCLUSION

We presented a novel refinement-based approach for effectively combining overapproximation techniques used by incomplete verifiers with linear-programming-based methods used in complete verifiers. We implemented our method in a system called *RefineZono* and showed its effectiveness on verification tasks involving feedforward and convolutional neural networks with ReLU activations.

Our evaluation demonstrates that *RefineZono* can certify robustness properties beyond the reach of existing state-of-the-art complete verifiers (these can fail due to scalability issues) while simultaneously improving on the precision of existing incomplete verifiers (which can fail due to using too coarse of an overapproximation).

Overall, we believe combining the strengths of overapproximation methods with those of mixed integer linear programming as done in this work is a promising direction for further advancing the state-of-the-art in neural network verification.

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
