# OpenReview forum: "Boosting Robustness Certification of Neural Networks"
_ICLR.cc/2019/Conference_

### Official Review · AnonReviewer3 · 2018-11-02
**a refined approach for robust verification, but experimental part could be stronger**

**Rating:** 4
**Confidence:** 3

**Review:**

This paper introduces a verifier that obtains improvement on both the precision of the incomplete verifiers and the scalability of the complete verifiers. The proposed approaches combines over-parameterization, mixed integer linear programming, and linear programming relaxation.

This paper is well written and well organized. I like the simple example exposed in section 2, which is a friendly start. However, I begun to lose track after that. As far as I can understand, the next section listed several techniques to be deployed. But I failed to see enough justification or reasoning why these techniques are important. My background is more theoretical, but I'm looking for theorems here, considering the complicatedness of neural network. All I am looking for is probably some high level explanation.

Empirically, the proposed approach is more robust while time consuming that the AI2 algorithm. However, the contribution and the importance of this paper still seems incremental to me.  I probably have grumbled too much about the lack of reasonings. As this paper is purely empirical, which is totally fine and could be valuable and influential as well.  In that case, I found the current experiment unsatisfying and would love to see more extensive experimental results.

---

> ### Author Response · Authors · 2018-11-24
> **Response to main question**
>
>
> Q1. My background is more theoretical, but I'm looking for theorems here, considering the complicatedness of the neural network. All I am looking for is probably some high-level explanation.
>
> R1. RefineAI is a new approach for proving the robustness of neural networks: it is more precise than current incomplete methods and more scalable than current complete methods. We believe this is a difficult problem and RefineAI is a promising step forward.
>
> Some key insights in the paper:
>
> Insight I: expensive but precise techniques like MILP solvers can be used for refinement earlier in the analysis but do not scale for refinement of neurons in later layers. However, they do substantially improve on incomplete verifiers.
>
> Insight II: not all neurons in the network contribute equally to the output and thus we do not need to refine all neurons in a layer. For this, we present a novel heuristic which improves the scalability of our approach while maintaining sufficient precision.

---

### Official Review · AnonReviewer1 · 2018-11-03
**Interesting idea but not fully evaluated**

**Rating:** 6
**Confidence:** 3

**Review:**

In the paper, the authors provide a new approach for verifying the robustness of deep neural networks that combines complete yet expensive methods based on mixed integer-linear programming (MILP) and incomplete yet cheap methods based on abstract interpretation or linear-programming relaxation. Roughly speaking, the approach is to run an abstract interpreter but to refine its results at early layers of a neural network using mixed integer-linear programming and some of later layers using linear programming. The unrefined results of the abstract interpreter help these refinement steps. They help prioritize or prune the refinement of the abstract-interpretation results at neurons at a layer. Using neural networks with 3, 5, 6, 9 layers and the MNIST dataset, the authors compared their approach with AI^2, which uses only abstract interpretation. This experimental comparison shows that the approach can prove the robustness of more examples for all of these networks.

I found the authors' way of combining complete techniques and incomplete techniques novel and interesting. They apply complete techniques in a prioritized manner, so that those techniques do not incur big performance penalties. However, I feel that more experimental justifications are needed. The approach in the paper applies MILP to the first few layers of a given network, without any further simplification or abstraction of the network. One possible implication of this is that this MILP-based refinement is applicable only for the first few layers of the network. Of course, prioritization and timeout of the authors help, but I am not sure that this is enough. Also, I think that more datasets and networks should be tried. The experiments in the paper with different networks for MNIST show the promise, but I feel that they are not enough.

* p3: Why is l6 = 0? I think that it is easy to figure out that max(0,x4) is at least 0.

* p4: [li,yi] for ===> [li,ui]

* p4: gamma_n(T^#_(x|->Ax+b)) ===> gamma_n(T^#_(x|->Ax+b)(a))

* p4: subseteq T^#...  ===> subseteq gamma_n(T^#...)

* p5: phi^(k)(x^(0)_1,...,x^(k-1)_p) ===> phi^(k)(x^(0)_1,...,x^k_p)

* p6: I couldn't understand your sentence "Note that the encoding ...". Explaining a bit more about how bounds computed in previous layers are used will be helpful.

* p6: I find your explanation on the way to compute the second ranking with weights confusing. Do you mean that your algorithm looks into the future layers of each neuron xi and adds the weights of edges in all the reachable paths from xi?

* p7: Why did you reduce epsilon from 0.07 to 0.02, 0.15 and 0.017?

---

> ### Author Response · Authors · 2018-11-24
> **Response to main questions**
>
>
> Q1. Is MILP-based refinement applicable only for the first few layers of the network?
>
> R1. Generally, such refinement is most effective in the initial layers: as the analysis proceeds deeper into the network, it becomes harder for the MILP solver to refine the bounds within the specified time limit of 1 second. This is due to the increase in the number of integer variables caused by the increase in the number of unstable units (as explained in the general section on unstable ReLU).
>
> Q2. Why is l6 = 0? I think that it is easy to figure out that max(0,x4) is at least 0.
>
> R2. We assume you mean l6=-0.5. The negative lower bound for x6 = ReLU(x4) is due to the Zonotope ReLU transformer shown in Figure 2 which permits negative values for the output.
>
> Q3. I couldn't understand your sentence "Note that the encoding ...". Explaining a bit more about how bounds computed in previous layers are used will be helpful.
>
> R3. We mean that both the set of constraints added by the LP encoding (Ehlers (2017)) and the Zonotope transformer (Figure 2) for approximating ReLU behaviour depends on the neuron bounds from the previous layers. The degree of imprecision introduced by these approximations can be reduced by propagating tighter bounds through the network.  We will clarify this.
>
> Q4. Do you mean that your algorithm looks into the future layers of each neuron xi and adds the weights of edges in all the reachable paths from xi?
>
> R4. Yes. We consider all outgoing edges from xi and add the absolute values of the corresponding weights.
>
> Q5. Why did you reduce epsilon from 0.07 to 0.02, 0.015 and 0.015?
>
> R5. The 5x100 network is trained using adversarial training and is thus more robust than the other networks which were not obtained through adversarial training. Thus, we chose a higher epsilon for it compared to the other networks (please see the comment in the general section on unstable ReLU).

---

### Official Review · AnonReviewer2 · 2018-11-04
**Interesting ideas but not persuasive enough**

**Rating:** 5
**Confidence:** 4

**Review:**

This paper proposed a mixed strategy to obtain better precision on robustness verifications of feed-forward neural networks with piecewise linear activation functions.

The topic of robustness verification is important. The paper is well-written and the overview example is nice and helpful.

The central idea of this paper is simple and the results can be expected: the authors combine several verification methods (the complete verifier MILP, the incomplete verifier LP and AI2) and thus achieve better precision compared with imcomplete verifiers while being more scalable than the complete verifiers. However, the verified networks are fairly small (1800 neurons) and it is not clear how good the performance is compared to other state-of-the-art complete/incomplete verifiers.

About experiments questions:
1. The experiments compare verified robustness with AI2 and show that RefineAI can verify more than AI2 at the expense of much more computation time (Figure 3). However, the problem here is how is RefineAI or AI2 compare with other complete and incomplete verifiers as described in  the second paragraph of introduction? The AI2 does not seem to have public available codes that readers can try out but for some complete and incomplete verifiers papers mentioned in the introductions,  I do find some public codes available:
* complete verifiers
1. Tjeng & Tedrake (2017): github.com/vtjeng/MIPVerify.jl
2. SMT Katz etal (2017): https://github.com/guykatzz/ReluplexCav2017

* incomplete verifiers
3. Weng etal (2018) : https://github.com/huanzhang12/CertifiedReLURobustness
4. Wong & Kolter (2018): http://github.com/locuslab/convex_adversarial

How does Refine AI proposed in this paper compare with the above four papers in terms of the verified robustness percentage on test set, the robustness bound (the epsilon in the paragraph Abstract Interpretation p.4) and the run time? The verified robustness percentage of Tjeng & Tedrake is reported but the robustness bound is not reported.  Also, can Refine AI scale to other datasets?

About other questions:
1. Can RefineAI handle only piece-wise linear activation functions? How about other activation functions, such as sigmoid? If so, what are the modifications to be made to handle other non-piece-wise linear activation functions?

2. In Sec 4, the Robustness properties paragraph. "The adversarial attack considered here is untargeted and therefore stronger than ...". The approaches in Weng etal (2018) and Tjeng & Tedrake (2017) seem to be able to handle the untargeted robustness as well?

3. In Sec 4, the Effect of neural selection heuristic paragraph. "Although the number of images verified change by only 3 %... produces tighter output bounds...". How tight the output bounds improved by the neuron selection heuristics?

---

> ### Author Response · Authors · 2018-11-24
> **Response to main questions**
>
>
> Q1. The verified robustness percentage of Tjeng & Tedrake is reported but the robustness bound is not.
>
> R1. The epsilon considered for this experiment is reported (page 7) and it is 0.03.
>
> Q2. Can RefineAI handle only piecewise linear activation functions? How about other activations such as sigmoid? If so, what modifications are needed?
>
> R2. RefineAI provides better approximations for ReLU because it uses tighter bounds returned by MILP/LP solvers. Similarly, we can refine DeepZ approximations for sigmoid (which already exist) by using better bounds from a tighter approximation, e.g., quadratic approximation.
>
> Q3. How is the verification problem affected by considering the untargeted attack as in this paper vs. the targeted attack in Weng et al (2018) and Tjeng & Tedrake (2017)?
>
> R3. Since the targeted attack is weaker, the complete verifier from Tjeng and Tedrake runs faster and the incomplete verifier from Weng et al. proves more properties in their respective evaluation than it would if they considered untargeted attacks as considered in this paper.
>
> Q4. How tight are the output bounds improved by the neuron selection heuristics?
>
> R4. We observed that the width of the interval for the correctly classified label is up to 37% smaller with our neuron selection heuristic.

---

### Author Response · Authors · 2018-11-24
**Answers to key questions**


We thank the reviewers for their feedback.

Below is a summary of key points, followed by further elaboration on each point. We also provide individual replies to each reviewer.

Summary points [short]

1. RefineAI is more precise than state-of-the-art incomplete verifiers.
2. RefineAI is more scalable than existing state-of-the-art complete verifiers, including the latest: https://openreview.net/forum?id=HyGIdiRqtm based on Tjeng & Tedrake.
3. RefineAI is applicable to much larger networks than shown in the paper.
4. Effectiveness of verification methods for neural networks is primarily affected by the number of unstable ReLU units, *not* by the number of neurons.
5. DeepZ [1], the domain used in our paper, is publicly available [3].

We are happy to provide further results or explanations if requested.

Summary points [longer]

→ RefineAI is more precise than all state-of-the-art incomplete verifiers.

This is because DeepZ has the same precision as Weng et. al (2018) and Kolter and Wong (2018) while being faster (unlike RefineAI, Weng et al. cannot handle convolutional nets). Then, based on DeepZ results, Refine AI computes more precise results.

→ RefineAI is more scalable than all state-of-the-art complete verifiers, including the latest: https://openreview.net/forum?id=HyGIdiRqtm, based on Tjeng & Tedrake.

This is because the above method uses Box to compute initial bounds and uses more expensive methods if required. Unfortunately, in deeper layers, Box analysis becomes too imprecise and does not help. As a result, the above approach primarily relies on LP to obtain tight bounds for formulating a MILP instance for the whole network. Determining bounds with LP for all neurons in the larger networks is prohibitive. For example, on the 9x200 network from our paper, determining bounds with LP for all neurons already takes > 20 minutes (without calling the MILP solver which is more expensive than LP) whereas DeepZ computes significantly more precise bounds than Box for deeper layers in few seconds.

This gives us considerably fewer candidates to refine using LP/MILP than the Box analysis provides. Note that Tjeng & Tedrake (2017) is in turn significantly faster than Reluplex.

→ RefineAI is applicable to much larger networks than shown in the paper.

We evaluated RefineAI on larger publicly available networks from [3]: three MNIST convolutional networks containing 3,604 (Conv1), 4,804 (Conv2), 34,688 (Conv3) neurons and one skip net containing 71,650 neurons. We also tried a CIFAR10 convolutional network with 4,852 neurons. As in the paper, we considered epsilon values for which the precision of DeepZ drops significantly. The performance numbers below show RefineAI scales to larger networks (DiffAI is a particular defense [2]):

          Dataset            Network       Epsilon     Adversarial training            Avg. runtime(s)
                                                                                                                       DeepZ       RefineAI
          MNIST	            Conv1              0.1            None                                  1.1            357
                                    Conv2              0.2            DiffAI                                  6.8           602
                                    Conv3              0.2            DiffAI                                     7          1011
                                    Skipnet          0.13            DiffAI                                 163           682
          CIFAR10           Conv            0.012            DiffAI                                  3.9           262

→ The effectiveness of a verification method for neural networks is primarily affected by the number of unstable ReLU units, *not* by the number of neurons.

This is because the speed of a complete verifier and the precision of an incomplete verifier are affected mainly by unstable ReLU units: those which can take both + and - values. Indeed, the speed of the MILP solver used in both RefineAI and the method based on Tjeng & Tedrake (2017) is adversely affected by the presolve approximations for such unstable units.

This explains why defending a network (e.g., via DiffAI) will make any verifier scale better (including RefineAI): because defended networks have much fewer unstable units than undefended networks.

[1] Fast and Effective Robustness Certification, NIPS’18
[2] Differentiable Abstract Interpretation for Provably Robust Neural Networks, ICML’18.
[3] DeepZ analysis: https://github.com/eth-sri/eran.

---

### Meta-Review · Area_Chair1 · 2018-12-19
**A novel and scalable approach to robustness analysis of neural nets**

**Confidence:** 4
**Recommendation:** Accept (Poster)

**Metareview:**

The paper addresess an important problem of neural net robustness verification, and presents a novel approach outperforming state of art; author provided details rebuttals which clarified their contributions over the state of art and highlighted scalability; this work appears to be a solid and useful contribution to the field.